# Usefulness of Saline Sealing in Preventing Pneumothorax after CT-Guided Biopsies of the Lung

**DOI:** 10.3390/diagnostics13233546

**Published:** 2023-11-28

**Authors:** Andrei Roman, Andreea Brozba, Alexandru Necula, Delia Doris Muntean, Paul Kubelac, Zsolt Fekete, Ciprian Tomuleasa, Csaba Csutak, Diana Feier, Roxana Pintican, Catalin Vlad

**Affiliations:** 1Faculty of Medicine, “Iuliu Hatieganu” University of Medicine and Pharmacy, 400347 Cluj-Napoca, Romania; andrei.roman678@gmail.com (A.R.); alexnecula10@gmail.com (A.N.); paulkubelac@gmail.com (P.K.); drfekete@gmail.com (Z.F.); ciprian.tomuleasa@gmail.com (C.T.); csutakcsaba@yahoo.com (C.C.); diana.feier@gmail.com (D.F.); roxana.pintican@gmail.com (R.P.); catalinvlad@yahoo.it (C.V.); 2Department of Radiology, Oncology Institute “Prof. Dr. Ion Chiricuta”, 400015 Cluj-Napoca, Romania; 3Department of Radiology, Cluj County Emergency Hospital, 400347 Cluj-Napoca, Romania; 4Department of Oncology, Oncology Institute “Prof. Dr. Ion Chiricuta”, 400015 Cluj-Napoca, Romania; 5Department of Radiotherapy, Oncology Institute “Prof. Dr. Ion Chiricuta”, 400015 Cluj-Napoca, Romania; 6Department of Hematology, Oncology Institute “Prof. Dr. Ion Chiricuta”, 400015 Cluj-Napoca, Romania; 7Department of Surgery, Oncology Institute “Prof. Dr. Ion Chiricuta”, 400015 Cluj-Napoca, Romania

**Keywords:** saline sealing, CT-guided lung biopsy, pneumothorax prevention

## Abstract

This study aimed to assess the effectiveness of saline sealing in reducing the incidence of pneumothorax after a CT-guided lung biopsy. This was a retrospective case-control study of patients who underwent CT-guided biopsies for lung tumors using 18 G semiautomatic core needles in conjunction with 17 G coaxial needles. The patients were divided into two consecutive groups: a historical Group A (n = 111), who did not receive saline sealing, and Group B (n = 87), who received saline sealing. In Group B, NaCl 0.9% was injected through the coaxial needle upon its removal. The incidence of pneumothorax and chest tube insertion was compared between the two groups. Multivariate logistic regression was performed to verify the contribution of other pneumothorax risk factors. The study included 198 patients, with 111 in Group A and 87 in Group B. There was a significantly (*p* = 0.02) higher pneumothorax rate in Group A (35.1%, n = 39) compared to Group B (20.7%, n = 18). The difference regarding chest tube insertion was not significant (*p* = 0.1), despite a tendency towards more insertions in Group A (5.4%, n = 6), compared to Group B (1.1%, n = 1). Among the risk factors for pneumothorax, only the presence of emphysema (OR = 3.5, *p* = 0.0007) and belonging to Group A (OR = 2.2, *p* = 0.02) were significant. Saline sealing of the needle tract after a CT-guided lung biopsy can significantly reduce the incidence of pneumothorax. This technique is safe, readily available, and inexpensive, and should be considered as a routine preventive measure during this procedure.

## 1. Introduction

Lung cancer is the most frequent type of cancer among men and the second most diagnosed cancer on the globe with 2,206,771 new cases in 2020 [1]. It is responsible for 18% of all cancer mortality with 1,796,144 deaths attributed to it in 2020, which makes it the leading cause of cancer-related deaths [1]. In the wake of new lung cancer screening strategies, the discovery of an increasing number of early-stage lung tumors can be expected [2,3]. According to the European Society of Medical Oncology guidelines, an attempt to obtain a pathological sample should be made for any suspicious lung nodule prior to surgery or stereotactic radiation therapy if the patient’s clinical condition and the nodule’s size or location allows it [4]. For advanced disease, where a surgical sample is unlikely to be obtained, the demand for adequate biopsy material needed to enable accurate molecular profiling has become imperative due to the availability of numerous targeted therapies [5]. Computed tomography-guided lung biopsy is an essential technique for characterizing pulmonary nodules.

Although fine needle aspiration can be used for the biopsy of lung tumors, accurate histopathological and molecular analysis is favored by larger tissue samples that can be obtained through core needle biopsy, especially in the absence of a cytopathologist on the site to ensure the adequacy of the sample [6,7,8,9,10]. The thicker needles that are used for core biopsies are, however, associated with a higher complication rate [11,12,13,14]. According to a recent meta-analysis, the overall complication rate for a CT-guided core lung biopsy is 38.8%, of which pneumothorax is the most frequent, occurring in 25.3% of cases, 5.6% requiring the insertion of a chest tube [15,16].

Various techniques have been attempted for reducing the risk of pneumothorax and chest tube insertion. These include maneuvers such as rapid roll-over, deep expiration and breath-hold, the injection of various sealant materials such as autologous clotted or non-clotted blood, collagen, haemocoagulase, or hydrogel plugs [17,18,19,20,21,22,23,24]. Saline sealing of the needle tract is another method that has been shown to be useful in reducing the rate of periprocedural pneumothorax [25,26,27,28].

Saline sealing is performed in conjunction with the coaxial biopsy technique by injecting NaCl 0.9% through the coaxial needle as it is removed at the end of the procedure. Saline solution has the advantage of causing no adverse reactions, being readily available and inexpensive. However, this technique has been described in fewer reports compared to the other pneumothorax prevention methods and evidence for its effectiveness is still limited.

Our study aimed to assess the effectiveness of the saline sealing technique in preventing pneumothorax using as a control group a consecutive series of patients on whom no specific pneumothorax prevention method was employed.

## 2. Materials and Methods

### 2.1. Patients

This was a retrospective case-control study and was approved by the institutional ethics committee of the Oncology Institute “Prof. Dr. Ion Chiricuta” with the approval protocol number 89/16.03/2021. Patients that underwent CT-guided biopsies of lung tumors were split into two groups. Group A, considered as the control group, consisted of a series of consecutive patients (August 2017–October 2020) who underwent biopsies without any sealing technique. Group B consisted of a series of consecutive patients (November 2020–January 2023) that underwent biopsies followed by saline sealing. Exclusion criteria for both groups were as follows: (a) the needle did not cross the aerated lung; (b) a hemorrhage was visible along the needle tract prior to its removal; (c) the procedure was aborted before the biopsy was taken; (d) the pneumothorax occurred prior to the removal of the coaxial needle; (e) a missing follow-up radiograph.

### 2.2. Procedure

The patients were referred for percutaneous biopsy by oncologists, pneumologists and thoracic surgeons after bronchoscopy was deemed to be either inconclusive or unfeasible due to the location of the tumor. If the tumor was in extensive contact with the chest wall, or more accessible metastases, such as cervical lymph nodes, hepatic or superficial lytic bone metastases were present, an ultrasound guided biopsy was performed. Informed consent was received from all patients prior to the procedure, including procedure-related complications and the administration of saline solution for Group B. Anticoagulant and antiplatelet therapy was ceased according to the Consensus Guidelines for Periprocedural Management of Coagulation Status and Hemostasis Risk in Percutaneous Image-guided Interventions and the coagulation status was verified on the day of the procedure [29]. The procedures were performed by a single radiologist using a 64 slice Optima 660 (GE Medical Systems, Chicago, IL, USA) CT scanner or a 64 slice Somatom Confidence (Siemens, Munich, Germany) CT scanner. An unenhanced low-dose (120 KV; 60 mA) planning CT scan of the thorax was performed prior to the biopsy and the shortest and safest puncture path was chosen, avoiding fissures and significant blood vessels. When possible, a prone position of the patient was preferred, as well as a perpendicular angle between the pleura and the needle. No sedation medication was used. The patient was instructed to maintain regular breathing and not to talk or cough during the procedure. The puncture site was disinfected and covered in sterile drapes, followed by local anesthesia using 1% Lidocaine. A 17 G coaxial needle was inserted in a stepwise manner towards the margin of the tumor using ultra-low-dose (120 kV; 20 mA) single- or triple-slice scans for guidance. The radiologist was present in the CT room during the entire procedure, the scans being started using a footswitch. No breathing commands were issued during the procedure. For tumors located in the lower lobes that were highly mobile during respiration, the scans were synchronized with the respiratory phase by observing the patient’s respiratory motion. The needle position was stabilized using gauzes soaked in povidone iodine. A semiautomatic, 18 G biopsy cutting needle (VELOX 2, Medax Medical Devices, Poggio Rusco, Italy) was inserted through the coaxial needle, obtaining 1–5 tissue samples that were stored in formalin solution. In patients belonging to Group A, the coaxial needle was removed after tissue sampling without any further action. In patients belonging to Group B, 3–5 mL of saline solution (NaCl 0.9%) was gradually injected through the coaxial needle during its removal at the end of the biopsy (Figure 1). The presence of pneumothorax and further complications were verified on a single-slice CT scan immediately following the procedure and at 2 h on a chest X-ray. If no pneumothorax was present and the patients were asymptomatic, they were released on the following day. If a small, asymptomatic pneumothorax occurred, patients received a second X-ray 2 h later and were released the following day if the pneumothorax remained relatively unchanged. A chest tube was inserted if the pneumothorax was progressing rapidly or was symptomatic by a surgeon who was blinded regarding the usage of saline sealing. The tube was removed 1–4 days later, after air leakage ceased completely.

### 2.3. Study Design

CT and X-ray images were retrospectively reviewed. Based on the CT images and the patient’s medical record, gender, age, tumor size, location, biopsy tract length, and the histopathological result were noted. The presence of emphysema was registered according to the Fleischner Society criteria [30]. The presence of a pneumothorax and its size were recorded based on the X-ray where it appeared largest. The insertion of a chest tube was recorded from the patient’s file.

The statistical analysis was performed using a GraphPad Prism 9.1.2 (GraphPad Software, Inc., San Diego, CA, USA). Data normality was verified using the Shapiro–Wilk test. The data were considered to be normally distributed for a *p* value > 0.05. To compare the two groups, a Fisher exact test was used for categorical variables and a Mann–Whitney test for continuous variables, not normally distributed. Multiple logistic regression was used in order to analyze the pneumothorax risk factors (gender, age, belonging to Group A, emphysema, tumor location, tumor size, biopsy tract length) and odds ratios were calculated. A *p* value < 0.05 was considered statistically significant.

## 3. Results

A total of 198 patients were included in the study, of whom 111 belonged to Group A and 87 belonged to Group B. The overall success rate was 97.4% (n = 193), with 2.6% (n = 5) inconclusive biopsies due to necrosis. The most prevalent type of cancer identified was adenocarcinoma, accounting for nearly 50% of all histopathological findings. Metastatic lung tumors were the second most common outcome, representing over 20% of all results (Table 1).

There were no significant differences between the two groups regarding age, number of tissue samples taken, the presence of emphysema, tumor location and biopsy tract length. Moderate centrilobular emphysema was present in three patients belonging to Group A and in two patients belonging to Group B. All other patients where emphysema was present had either a trace or mild centrilobular or paraseptal emphysema. In Group A (37.8%, n = 42), there was a significantly lower (*p* = 0.03) proportion of female patients, compared to Group B (54.1%, n = 47). The average nodule size was significantly larger (*p* = 0.04) in Group A (32.7 ± 16.4), compared to Group B (29.2 ± 17.5) (Table 2).

Group A (35.1%, n = 39) had a significantly higher (*p* = 0.02) pneumothorax rate compared to Group B (20.7%, n = 18) (Figure 2). If pneumothorax was present, its thickness was not significantly different (*p* = 0.6) between the two groups. The insertion of a chest tube was required in 5.4% (n = 6) of patients belonging to Group A and in 1.1% (n = 1) of patients belonging to Group B (Figure 3); however, the difference was not statistically significant (*p* = 0.1).

Multiple logistic regression showed that emphysema (OR = 3.5, *p* = 0.0007) and belonging to Group A (OR = 2.2, *p* = 0.02) represented a significant independent risk factor for pneumothorax. Gender, age, tumor location, tumor size and biopsy tract length had no significant influence on the occurrence of pneumothorax (Table 3). Multiple logistic regression could not be performed for the analysis of a chest tube insertion due to the limited number of events.

Other complications besides pneumothorax were small hemothorax (1.01%, n = 2), severe, but self-limiting hemoptysis (1.01%, n = 2), a large reactive pleural effusion (0.51%, n = 1), and a necrotic tumor infection with sepsis (0.51%, n = 1). Mild intra- or periprocedural hemoptysis was not recorded in the patient’s file. No deaths occurred consecutive to the biopsies.

## 4. Discussion

This study shows that the saline sealing technique significantly diminishes pneumothorax risk after CT-guided biopsies, with a tendency towards reducing the chest tube insertion rate. In the analyzed patient group, the only other independent factor significantly influencing the pneumothorax rate was the presence of emphysema.

Pneumothorax is the most common complication associated with CT-guided biopsies of the lung occurring in 25.3% of patients. Despite the fact that most instances of pneumothorax are asymptomatic and self-limiting, in approximately 5.6% of cases drainage is needed [15]. In order to reduce the morbidity and costs associated with pneumothorax, various preventive methods have been proposed.

One group of these is related to patient positioning or respiratory maneuvers such as positioning the patient biopsy side down during the procedure, rapid or slow roll-over to puncture the site down after the biopsy, or breath hold on needle extraction [24,31]. The studies that compared the benefit of rapid roll-over vs. slow or no roll-over to puncture the side down after the lung biopsy showed no significant reduction in pneumothorax incidence; however, the rate of drainage catheter placement due to pneumothorax was significantly lower in the treatment group with an overall incidence of 1.9% compared to 5.2% in the control group [18,24,32]. Drumm et al. assessed the effectiveness of positioning the patient with the puncture side down during and immediately after the CT-guided lung biopsy and showed that the pneumothorax rate was significantly reduced compared to a supine or prone position (10.6% compared to 27.2%), but that no substantial difference in chest drain insertion was reported [31]. Another study evaluated the benefits of breath-hold after forced expiration before extracting the needle, demonstrating a statistically significant reduction in the pneumothorax rate (8.2% compared to 15.8%), but no significant reduction in the number of patients requiring a drainage catheter [33]. Although altering the patient’s position seems promising, adjusting and maintaining a certain attitude may be difficult for a postoperative patient to achieve.

The other group of preventive measures consists of injecting a sealing substance along the needle tract such as a saline solution, clotted or non-clotted autologous blood, and heterologous plugs consisting of collagen foam, fibrin glue, hydrogel plug or gelatin powder [17,18,19,21,22,23]. The sealing techniques are based on the observation that a pneumothorax occurs through the parenchymal and pleural defect that remains after the removal of the needle. Filling that defect and the nearby alveoli using a sealant is presumed to stop the airflow towards the pleural space. A multitude of reports regarding the efficiency of the blood patch technique and of various heterologous plugs exist; however, these techniques have some downsides. Although inexpensive, and theoretically devoid of side-effects, the blood patch technique can be time-consuming in patients with difficult venous access, or with inadequate cannulation, thus prolonging the discomfort of the procedure [34]. Heterologous plugs can be expensive, their preparation can be time-consuming, and some of them have been shown to cause a granulomatous inflammatory reaction [35,36].

Six research articles included in the meta-analysis by Huo et al. using autologous blood patches showed an overall reduction in the risk of pneumothorax (27.9% compared to 40.1% in the control group) and of a chest drain insertion (4.8% compared to 11.1%) [18]. Lang et al. used a technique in which the clotted blood was separated and the supernatant was injected into the track and at the level of the biopsied nodule, and the solid clot elements were deployed at the periphery [17]. In a retrospective analysis performed by Perl and colleagues, intraparenchymal blood patching reduced the incidence of pneumothorax for nodules located in the lower lobes, closer to the pleura (<2 cm) or deeper inside the lung (>4 cm) and when four or more samples were taken [22]. Zlevor et al. studied the effects of blood patching as a preventive measure as well as a therapeutic method for pneumothorax to avoid a chest tube placement and reported a success rate of 83.4% [21].

The most obvious theoretical disadvantages of saline sealing compared to other sealing techniques are the rapid resorption and diffusion of saline solution in the lung parenchyma and the fact that it may be more easily pushed out from the needle tract by the pulmonary pressure due to its low viscosity. Possibly due to these assumptions, the efficacy of saline sealing has been less extensively investigated. However, the saline sealing technique has the advantage of being both inexpensive and easily applied, without having any side-effects, and its effectiveness has been shown to hold promise based on the few existing reports (Table 4).

The British Thoracic Society (BTS) recommends complication rates equal to or below 20.5% for pneumothorax and 3.1% for pneumothorax requiring chest drainage [37]. With a pneumothorax rate of 35.1% and a chest tube insertion rate of 5.4%, Group A did not comply with the BTS quality requirements. By applying the saline sealing technique, the pneumothorax and chest tube insertion rates dropped to 20.7% (just above BTS standards) and 1.1% (within BTS standards), respectively. In our study, the decrease in the frequency of pneumothorax was statistically significant, and comparable to the frequency reported by Babu et al., but less important compared to the decrease reported by Billich et al. and Li et al. [25,26,28]. All previous studies reported a significant decrease in the rate of drainage insertion. A decrease was also noticeable in our study; however, the difference was not statistically significant, likely due to the relatively low number of patients and of chest tube insertions in the control group. It is worth mentioning that all the above-mentioned studies had higher chest tube insertion rates in the control groups, which suggests possible differences regarding patient selection, technique, or the threshold for inserting a chest tube, the latter being the most likely explanation, as their pneumothorax rates were lower than in our study with the exception of Babu et al. (Table 4).

In contrast with the study of Billich et al., patients with tumors in direct contact with the chest wall were excluded as air leakage from the lungs is highly unlikely in this context [25]. Unlike in the previously mentioned papers, patients who had visible hemorrhage along the needle tract prior to its removal were excluded, as this has been shown to be associated with a lesser pneumothorax rate [38,39].

After correcting for other known pneumothorax risk factors using multiple logistic regression, besides the presence of emphysema (OR = 3.5), only the saline sealing technique (OR = 1/2.26 = 0.44) showed a statistical significance. The results were equivalent or favorable compared to the results published in the meta-analysis by Huo et al. for the positional and breathing techniques (OR = 0.48–0.69) as well as for the blood patch technique (OR = 0.57) and the heterologous plug technique (OR = 0.47) [18]. It must be mentioned that some of the studies included in the above-mentioned meta-analysis are older than 30 years, and also include fluoroscopic guidance. In addition, most biopsies were performed using 19 G coaxial needles and 20–22 G biopsy needles, which makes an accurate comparison difficult.

There are a few limitations to the present study, including its retrospective nature, the single center approach, and the relatively low number of patients. There was a difference between the groups regarding gender and tumor size, with significantly more female patients and slightly smaller tumors (29.2 mm vs. 32.7 mm) in the saline sealing group, but neither of these factors showed a significant influence on the occurrence of pneumothorax at multiple logistic regression. Nevertheless, a smaller tumor size has been shown to be a risk factor for pneumothorax and may have influenced the results [14]. Another source of bias could be the operator’s experience, as the biopsies belonging to Group A were all performed prior to those in Group B, despite the fact that the overall technique remained identical. The lack of standardization regarding the quantity of saline that was injected could be another confounding factor for this study. It is possible that injecting a larger quantity of saline solution would further reduce the risk of pneumothorax, without any significant side-effects.

## 5. Conclusions

Besides the presence of emphysema, the only independent factor influencing the occurrence of pneumothorax was the application of the saline sealing technique. Our study shows that using saline solution to seal the needle tract after a percutaneous pulmonary biopsy significantly reduces the pneumothorax rate and shows a tendency towards a reduction in the drainage insertion rate. Further randomized controlled trials could prove useful in determining the most appropriate needle tract sealing technique.

## Figures and Tables

**Figure 1 diagnostics-13-03546-f001:**
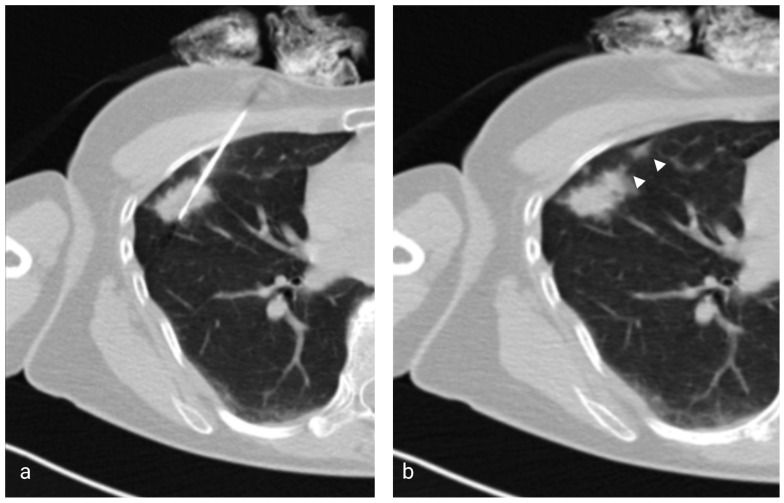
(**a**) Axial low-dose CT showing the 17G coaxial needle and the 18G biopsy needle within a right middle lobe mass. (**b**) Image taken immediately after the injection of saline solution and removal of the coaxial needle showing the hyperattenuating needle tract (arrowheads).

**Figure 2 diagnostics-13-03546-f002:**
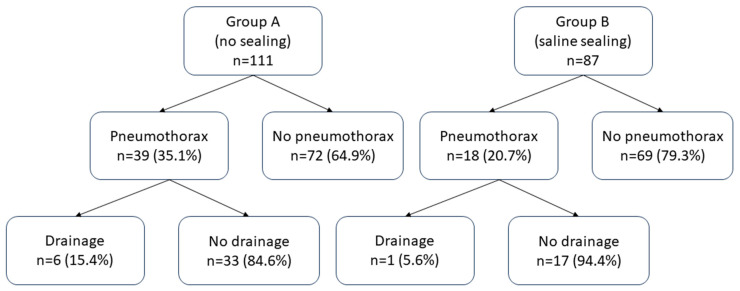
Proportions of pneumothorax and drainage insertion among the two groups.

**Figure 3 diagnostics-13-03546-f003:**
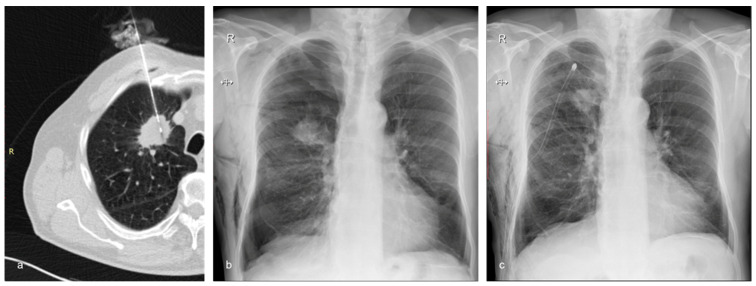
A 72-year-old man with emphysema and a tumor in the right upper lobe belonging to Group B (the treatment group). The nodule was confirmed to be adenocarcinoma by the histopathological examination. (**a**) Axial CT showing the 17G coaxial needle and the 18G biopsy needle within the lung mass. (**b**) Chest X-ray performed 2 h after the biopsy showing moderate post-procedural pneumothorax and subcutaneous emphysema. The patient complained of right chest pain and dyspnea. It was decided to insert a chest tube to prevent further expansion of the pneumothorax and to alleviate the symptoms. (**c**) Chest X-ray of the same patient following the insertion of the chest drain.

**Table 1 diagnostics-13-03546-t001:** Histopathological results.

Histopathological Result	Number of Histopathological Results (%)
**NSCLC**	
Adenocarcinoma	96 (48.48%)
Squamous cell carcinoma	15 (7.58%)
NSCLC-NOS	12 (6.06%)
**Neuroendocrine neoplasms**	
SCLC	2 (1.01%)
Neuroendocrine tumor-NOS	11 (5.56%)
**Metastases**	40 (20.20%)
**Benign (fibrosis, inflammation)**	11 (5.56%)
**Necrosis**	5 (2.53%)
**Other tumors**	
Mesenchymal tumors specific to the lung (Pulmonary hamartoma)	2 (1.01%)
Malignant fibrous histiocytoma	1 (0.51%)
Inflammatory myofibroblastic tumor	1 (0.51%)
Hematolymphoid tumors (Large B-cell lymphoma)	2 (1.01%)

NSCLC—Non-Small Cell Lung Cancer. NOS—Not Otherwise Specified. SCLC—Small Cell Lung Cancer.

**Table 2 diagnostics-13-03546-t002:** Clinical characteristics of patients.

Variable	Group A	Group B	*p*
**N**	111	87	
**Age (years)**			
Mean ± SD	63.7 ± 8.9	63.4 ± 9.0	0.8
Range	30–80	34–82	
Normal distribution	No	No	
**Gender**			
Men	69 (62.2%)	40 (45.9%)	0.03
Women	42 (37.8%)	47 (54.1%)	
**Biopsy fragments**			
Mean ± SD	2.1 ± 0.9	2.1 ± 0.7	0.5
Range	1–5	1–3	
Normal distribution	No	No	
**Emphysema**			
Yes	31 (27.9%)	31 (35.6%)	0.2
No	80 (72.1%)	56 (64.4%)	
**Nodule size (mm)**			
Mean ± SD	32.7 ± 16.4	29.2 ± 17.5	0.04
Range	9–87	8–110	
Normal distribution	No	No	
**Nodule location**			
Right lower lobe	38	15	
Right middle lobe	3	2	
Right upper lobe	28	35	
Left lower lobe	16	14	
Left upper lobe	26	21	
**Biopsy tract length (mm)**			
Mean ± SD	17.7 ± 10.4	19.2 ± 10.1	0.2
Range	2–54	4–43	
Normal distribution	No	No	
**Pneumothorax**			
Yes	39 (35.1%)	18 (20.7%)	0.02
No	72 (64.9%)	69 (79.3%)	
**Pneumothorax size (mm)**			
Mean ± SD	14.9 ± 16.1	13.3 ± 15.6	0.6
Range	2–61	3–56	
Normal distribution	No	No	
**Chest tube insertion**			
Yes	6 (5.4%)	1 (1.1%)	0.1
No	105 (94.6%)	86 (98.9%)	

Group A—control group. Group B—treatment group. N—number of patients included in the study. SD—standard deviation.

**Table 3 diagnostics-13-03546-t003:** Multiple logistic regression evaluating pneumothorax risk factors.

Variable	OR	95% CI	*p* Value
Gender (female)	0.52	0.24–1.07	0.08
Age	1.00	0.96–1.04	0.7
Group A	2.26	1.10–4.80	0.02
Emphysema	3.50	1.71–7.33	0.0007
Location (RLL)	Reference		
Location (LLL)	1.56	0.57–4.28	0.3
Location (ML)	1.18	0.12–9.14	0.8
Location (LUL)	0.43	0.15–1.16	0.1
Location (RUL)	0.61	0.24–1.48	0.2
Nodule size	0.99	0.97–1.01	0.4
Biopsy tract length	1.00	0.97–1.03	0.7

OR—Odds Ratio. CI—Confidence Interval. RLL—Right Lower Lobe. LLL—Left Upper Lobe. ML—Middle Lobe. LUL—Left Upper Lobe. RUL—Right Upper Lobe.

**Table 4 diagnostics-13-03546-t004:** Articles published on saline sealing.

Study	Method	N	Incidence of PTX	*p*	Chest Drainage	*p*
Billich 2008 [25]	NaCl 0.9%	140	34% vs. 8%	<0.001	11.4% vs. 1.4%	0.01
Li 2015 [26]	NaCl 0.9%	322	26.1% vs. 6.2%	<0.001	5.6% vs. 0.6%	0.01
Khorochkov 2018 [27]	NaCl 0.9% and rapid patient roll-over	278	25% vs. 20%	0.22	10% vs. 3.9%	0.03
Babu 2020 [28]	NaCl 0.9%	200	46% vs. 32%	<0.05	7% vs. 1%	<0.05
The present study	NaCl 0.9%	198	35.1% vs. 20.7%	0.02	5.4% vs. 1.1%	0.1

## Data Availability

The data presented in this study are available on request from the corresponding author. The data are not publicly available due to institutional policy.

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
