# Peer review of "Usefulness of Saline Sealing in Preventing Pneumothorax after CT-Guided Biopsies of the Lung"

_diagnostics, 2023, doi:10.3390/diagnostics13233546_

Round 1

Reviewer 1 Report

Comments and Suggestions for Authors

The authors present a retrospective study that discusses techniques that can reduce the side effect of CT guided lung biopsies especially pneumothorax using the saline sealing method through the coaxial needle which has been reported but is not as well investigated and used as other methods such as the rapid roll over, deep expiration, breath heal or sealing with autologous or heterozygous anticoagulants. The authors discuss the advantages and disadvantages and compare their finding to those in literature. The findings are well presented.

A question for the authors.

There was a statistical difference in the gender for the two groups that were asssesd for the complication using the saline sealant and those that did not but beyond the table this was not mentioned or discussed. Could this have an impact of the conclusions drawn from the study?

Author Response

We want to thank the reviewer for the favourable feedback offered to our work.

We have also tried to find an explanation for the higher prevalence of women in group B. This could be partially explained by the higher proportion of metastases originating from gynecological malignancies biopsied in group B (73%), compared to group A (56%), but chance is likely to have also played a role. We have mentioned the gender disparity in the "limitations" section of the sudy: "There was a difference between the groups regarding gender and tumor size, with significantly more female patients and slightly smaller tumors (29.2 mm vs. 32.7 mm) in the saline sealing group, but neither of these factors showed a significant influence on the occurrence of pneumothorax at multiple logistic regression."

Also, there is no data in the literature suggesting that gender might influence pneumothorax rates.

Reviewer 2 Report

Comments and Suggestions for Authors

1. In the original study titled "Usefulness of saline sealing in preventing pneumothorax after CT-guided biopsies of the lung", the authors provide a rationale for using saline sealing to prevent lung biopsy complications. The main study strengths are its detailed methodology and high relevance. The main study weaknesses are related to design, sample estimation, statistical analysis and effect strength.
2. The study is relevant to an audience of radiologists and thoracic surgeons.
3. The conclusions are only partly consistent with the evidence provided, as other sealing techniques nor economic efficiency have not been evaluated with the current sample.
4. The figures and tables are informative, requiring no further revision.

Consider further improving the manuscript's quality by:
- providing additional rationale as to why the study was retrospective, when a novel sealing method using saline solution had been employed (i.e., prospective study);
- estimating the required sample using https://shiny.ctu.unibe.ch/presize/ or similar tool due to prospective study design and commenting whether the sample was adequate for statistical testing with high reproducibility;
- performing the data normality check via Shapiro-Wilk test and including its results;
- mentioning to the reader that observed differences between tumor size in two groups had been statistically significant with only small effect size (i.e., several millimeters) as it may impact the study results;
- beginning the "Discussion" section with a sentence briefly summarizing the chief study finding.

Round 2

Reviewer 2 Report

Comments and Suggestions for Authors

The authors have provided sufficient responses to reviewer's queries, improving the manuscript.

Author Response

Thank you for the help in improving our work!